# The Contribution of Educational Psychology to South African Preservice Teacher Training and Learner Support

Motlalepule Ruth Mampane 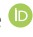

Department of Educational Psychology, University of Pretoria, Pretoria 0002, South Africa;
ruth.mampane@up.ac.za

**Abstract:** Teacher education programmes are developed around the theoretical and practical understanding of child development, learning, assessment, behaviour management and motivation, which are areas of expertise in educational psychology. This paper aims to (a) critically investigate the contribution of educational psychology in the training of preservice teachers at South African universities and (b) understand the distribution of educational psychologists in public schools to support teaching and learning. A narrative literature review and email requests for unpublished documents from four educational psychologists were used as methods to collect literature in order to answer the following questions: What contribution does educational psychology make to training preservice teachers at public universities in South Africa? What contributions do educational psychologists make to support learners in South African public schools? Analysis was carried out by identifying recurring patterns in the literature reviewed. This study found that of the 26 public universities in South Africa, there are only 6 universities that offer educational psychology programmes. Educational psychology programmes in higher education institutions are in decline, leading to a decrease in the number of qualified educational psychologists. This decline negatively affects the involvement of educational psychologists in training preservice teachers in educational psychology modules or courses. Therefore, the inclusion of educational psychology as a core or fundamental module in the curriculum of preservice teachers to avoid dependence on the decreasing number of educational psychologists in higher education institutions is key. An increase in teacher training programmes in higher education should be merged with an equal increase in educational psychology core or fundamental courses in the curriculum of preservice teachers.

**Keywords:** educational psychology; preservice teacher; child development; high education; department of basic education; learning support; Health Professions Council of South Africa

## 1. Introduction

Educational psychology informs preservice teachers on how learning takes place and how an individual learns. The literature on the role of educational psychology in teacher training programmes can be traced back to educational psychologists such as James [1] and Thorndike [2]. The author of [3] confirms that educational psychology is the study of child development and learning, counselling, guidance, special education and reading. The training of preservice teachers focuses on, e.g., the learning process and learning theories, including different domains of child development like cognitive, emotional, social and physical development [1,2]. Focusing on theories of child development and learning and how they contribute to teaching and learning helps us to understand the learning process. Educational psychology's role in training preservice teachers aims to equip teachers with the necessary knowledge and skills to understand, identify and assess learning problems using theories of development and learning and non-standardised tests, as well as to support or refer learners for further support. Furthermore, preservice teachers can be exposed to and trained in using and developing non-standardised assessment tests that are essential to teaching and learning.

Education and educational psychology are interdependent [4]. The authors of [5] (p. 73) define educational psychology as "the development and application of psychological principles to education, as well as the adoption of psychological perspectives on education". This definition draws a link between educational psychology and education. The American Psychology Association Dictionary defines educational psychology as "a branch of psychology dealing with the application of psychological principles and theories to a broad spectrum of teaching, training, and learning issues in educational settings. Educational psychology also addresses psychological problems that can arise in educational systems" [6]. The Health Professions Council of South Africa (HPCSA) defines educational psychology as a "specialist category within professional psychology that promotes the learning, academic performance, and the behavioural, social and emotional development of learners of all ages, with an emphasis on children and young people, in the context of schools and other educational settings" [7]. The APA and HPCSA definitions acknowledge the interactive relationship between the science of psychology and the profession of teaching. Historically, the literature acknowledges the significant role of educational psychologists in training preservice teachers [2,8–10].

Psychological theories on human development, cognitive functioning, learning, assessment, motivation, classroom management and socioemotional domains [10,11] form the basis of many educational programmes and guide education policies. Education policies like the South African Schools Act (SASA) [12] inform caregivers and parents about school readiness by spelling out the expected developmental milestones and provide significant direction to stakeholders for understanding children's maturation throughout the early and later school years [13,14]. Knowledge of the expected developmental milestones enables the early detection of developmental and learning barriers that require early intervention, inclusion and support in the education system [14]. The theories of psychologists such as Maslow, Piaget and Vygotsky [15,16] highlight normal and abnormal human development and can facilitate early problem detection and support.

Over a decade ago, Patrick et al. [10] identified three challenges for educational psychology in education systems. These challenges, which are still relevant today, include communicating the relevance of educational psychology in education, developing collaborative relationships with colleagues in education programmes, and documenting ways that educational psychology courses can make a difference in the practice of teachers [10] (p. 71). The author of [17] looked at the changes and developments of educational psychology in South Africa since 1994 and suggested that knowledge of developmental theories and how they influence the education system is insufficient in teacher education. Further research is needed to understand the role and contribution of educational psychology in teacher education and learner support. This paper seeks to critically interrogate the contribution of educational psychology in the training of preservice teachers in higher education institutions and learner support in basic education.

*Literature Review*

Educational psychology, a psychology discipline that specialises in learning and education, focuses (among others) on theories of learning, human development, behaviour, socioemotional development and cognition—all of which are keys to education systems. In South Africa, educational psychology was initially seen as a helping and intervention profession [17,18]. Teachers as subject specialists who rely on psychological theories to understand learning and teaching requirements per grade and developmental phase. In South Africa, children start formal schooling at Grade R when they are approximately five years old, with the expectation that this developmental age enables children to be physically, cognitively, socially and emotionally ready to enter and cope with the formal education system and the curriculum of the grade. Since Grade R is formal education, it is essential to ensure that before a child enters Grade R, he/she is ready to learn at school to avoid exposing the child to learning problems. School readiness focuses on children's development and their ability to self-regulate. The following domains of development are

key to assessing school readiness: physical development, academic readiness, cognitive skills and socioemotional functioning of the child being assessed [10]. A teacher, a school counsellor or an educational psychologist can carry out the school readiness assessment. Institutions of higher learning, where educational psychologists are trained, need to move beyond the traditional focus on theoretical knowledge of human development, learning and behaviour and pursue specific psychological core competencies that are essential for student teachers to understand and work with learners in schools.

Teacher education planning requires academics in the discipline of education to take the lead in the curriculum and knowledge development of student teachers. Even though teachers are not psychologists, they require significant knowledge and understanding of human development and learning. The authors of [9] propose that psychological topics on learning, child development and assessment should be included in teacher education curricula at the university level. This paper argues that educational psychology has a crucial role in teacher education. The educational psychology programme in South Africa is the only one that trains educational psychologists. Similarly, higher education institutions with educational psychology departments employ educational psychologists to train preservice teachers and educational psychology students. With a master's degree in educational psychology, students can qualify to be educational psychologists after meeting all the requirements for the degree and profession. Thus, the connection between educational psychology departments, programmes and educational psychologists training preservice teachers in higher education institutions can be interdependent. Higher education (HE) institutions focus on training preservice teachers, and educational psychology programmes contribute to the training. The Department of Basic Education in South Africa employs qualified (trained) teachers and qualified educational psychologists. Thus, it is assumed that professional teachers and educational psychologists focus on learners at the basic education level.

The author of [19] reported that there are 48 universities in South Africa (22 private and 26 public), of which only 13 (12 public and 1 private) offer HPCSA-accredited educational psychology programmes at the masters and honours levels. A total of 6 public universities offer an HPCSA-accredited master's in educational psychology programmes, and 12 offer a BPsych (honours) degree [20]. The decline in the number of higher education institutions that train educational psychologists will ultimately affect the distribution of educational psychologists to schools and higher education institutions to train preservice teachers and support learners in schools.

The problem in South Africa is that the Department of Basic Education has not prioritised the employment of educational psychologists in public schools to support learning. The responsibility of employing educational psychologists in schools is devolved to School Governing Bodies (SGBs); thus, only schools that can afford such services can employ the services of an educational psychologist. Hence, schools that do not have the necessary financial resources do not have such a professional as part of the school staff, and one of the subject teachers will have to identify, support and refer learners with learning barriers to psychologists who are at the district level as part of the district-based support team. The Education White Paper 6: Special Needs Education [21], which focuses on inclusive education, provides guidelines on how every school in South Africa can work towards early identification of learning barriers, support such learners and provide referral to psychologists at the district level when necessary. Teachers are expected to have full knowledge and understanding of how to implement the inclusive education policy through early identification of learning problems, assessment and support of learners experiencing learning problems. The authors of [9] propose that psychological topics on learning, child development and assessment should be included in university teacher education curricula to empower teachers with the necessary knowledge to assess, identify and support learners with learning problems.

The content of teacher education programmes aligns with the primary education curriculum to enable subject specialisation for teachers. Accordingly, [22] emphasises that

it is critical for teacher education programmes to have close relationships with schools. Teacher education programmes should have a strong coherence and integration among the courses or modules in their curriculum, provide supervised practical (work-integrated learning) work in schools, and enable the use of pedagogies that link theory and practice [22]. Besides specialised subjects, it is important for teacher programmes to include modules or courses in their curriculum that focus on classroom management, how to teach and how learning occurs. It is expected that such a curriculum for teacher training should cover what is suggested by its framework for understanding teaching and learning [23]. The framework for understanding teaching and learning proposed by [23] (pp. 11–12) outlines the following three knowledge areas as essential to teacher education:

1. Knowledge of language development and how learners learn and develop within social contexts;
2. Knowledge of the curriculum content and goals, subject matter, skills to be taught, disciplinary demands, student needs and the social purposes of education;
3. Understanding the skills for teaching and the content, pedagogical knowledge, and knowledge of teaching diverse learners, as well as understanding of how to conduct assessment and how to construct and manage a productive classroom.

This framework indicates that a preservice teacher's curriculum should include knowledge of child development, teaching diverse learners (inclusive education), and understanding assessment, among other components. Most of these requirements are part of educational psychology programmes (assessment, child development, and inclusive education). According to its Deputy Director General, Mweli (16 May 2023, National Seminar on reading literacy programme presentation), the following statistics are presented by the South African Department of Basic Education:

1. There are 24,871 schools in the nine provinces of South Africa, of which 22,589 are public schools, and 2282 are independent schools;
2. Of the nine provinces, there are 75 districts and 889 circuit offices;
3. The number of learners in public schools is 12,684,886, and the number of teachers (or educators) is 405,626;
4. Independent schools have 735,085 learners and 45,368 teachers;
5. Overall, South Africa has 13,419,971 learners and 405,993 teachers.

The above statistics indicate that not all learners in South African public schools can access the support of an educational psychologist when needed. However, if all qualified teachers' curricula included educational psychology in training, they would be well equipped to understand child development and learning, assessment, identification of learning problems, and how to support and refer learners with learning problems. According to [24] (p. 41), in August 2019, the Health Professions Council of South Africa (HPCSA) reported that South Africa had 1672 registered educational psychologists. The racial and gender distribution was reported as follows: 344 were black (177 black African, 59 coloured, and 103 Indian), 981 were white, and race was unknown for 63 individuals [24]. The information from Mweli on the number of public schools in South Africa and the HPCS information on the number of registered educational psychologists confirm that South Africa does not have a sufficient number of educational psychologists to support learners in schools. Thus, it is not possible for every South African child to have access to an educational psychologist during their school years. Based on the statistics mentioned above, the inclusion of educational psychology curriculum as a core or fundamental course in the education of preservice teachers should be prioritised and made mandatory.

## 2. Theoretical Framework

The author of [25] argues that critical theory is about emancipation, has a liberating influence, and enables a world that satisfies the needs and powers of human beings. In the view of [26], critical theory adopts a normative approach based on the judgment that domination is a problem and that a domination-free society is needed. The basic tenet

of critical theory is to analyse social conditions, criticise unjustified use of power, and "change established social traditions and institutions so that human beings are freed from dependency, subordination and suppression" [27]. The authors of [25] adds that critical theory research should satisfy the three criteria of explanatory, practical and normative separately as well as at the same time. This article discusses educational psychology's contributions to preservice teacher training and school learner support. In the process, it critically discusses areas of strengths and challenges that can impede the contribution of educational psychology to teacher education. The identified challenges might point out the limitations of current educational psychology programmes and how change can be achieved, and possibly provide achievable, practical goals for social transformation [25]. Even though critical theory aims to expose and critique the abuse of power, to give voice to the oppressed, and to liberate and discourage domination of the powerless by the powerful, this article shows that learners who cannot access educational psychology services remain powerless and voiceless. Therefore, the following question remains: how can they be assisted in order to gain access to such services?

## 3. Research Methods

Two data collection methods were used to generate data for this study. Firstly, a general narrative literature review was conducted to provide a review of the essential aspects of the topic under discussion, namely the contribution of educational psychology in training preservice teachers from a South African perspective. The authors of [28] state that a narrative literature review links studies on different topics to interconnect and reinterpret their research findings. This narrative review aims to survey current state of knowledge on educational psychology's role in preservice teachers' training by pulling together existing literature and integrating and interconnecting ideas.

A narrative literature review synthesises published literature on a topic and describes the topic's current state [29]. According to [30] (pp. 24–25), there are four types of narrative reviews:

1. General literature review provides a review of the most critical aspects of current knowledge of a topic;
2. Theoretical literature review examines how theory shapes or frames research;
3. Methodological literature review describes the research methods and design;
4. Historical literature review examines research for a particular period, starting with the first time an issue, concept, theory or phenomenon emerged in the literature, and then tracing its evolution within the scholarship of a discipline.

This study followed the general literature review to provide a review of critical aspects of the knowledge of educational psychology, teacher education and learning support in both higher and basic education systems in South Africa.

Secondly, an email was sent to four senior education specialists and qualified educational psychologists employed by the Department of Basic Education as part of the district-based support teams to request documents that define their roles, responsibilities and overall job descriptions within the Department of Basic Education and schools. The email was sent to education specialists in Tshwane, the Free State and Johannesburg, requesting access to existing guidelines, correspondence and circulars which clarify how educational psychologists are distributed. The lack of formal documents to clarify the distribution and allocation of educational psychologists to schools was reported, and email reports were used instead. Literature was analysed by identifying major relationships, patterns and arguments from the reviewed literature [28,29].

*Research Questions and Objectives*

The following research questions informed the discussions, the literature reviewed, and the conclusions reached:

1. What is the contribution of educational psychology to the training of preservice teachers at public universities in South Africa?

2.    What is the contribution of educational psychology to support learners in South African public schools?

The objectives of this study following a critical review of the literature on the contribution of educational psychology in the training of preservice teachers are:

1.    To understand the contribution of educational psychology to teacher education training in South African higher education institutions;
2.    To understand the role of educational psychologists in learner support.

## 4. Research Findings and Discussion

The literature reviewed presented the following patterns: identification of universities training educational psychologists and teachers (see Table 1). Universities train teachers only (see Table 2) and the conclusion is that, there is a decline in educational psychology programmes in South Africa and the motivation including educational psychology as a fundamental or core module or course for preservice teachers' curricula.

**Table 1.** List of universities that offer HPCSA-accredited master's programmes in educational psychology in South Africa.

| University | Province | Faculty Where the Department/School Resides | Department's Involvement in Teacher Education |
|---|---|---|---|
| University of Johannesburg | Gauteng | Education | Offering preservice qualifications and training in the Foundation, Intermediate, Senior and FET phases, as well as the PGCE. |
| University of KwaZulu-Natal (UKZN) | KwaZulu-Natal | Applied Human Sciences | Teaching in undergraduate modules |
| University of Pretoria (UP) | Gauteng | Education | Teaching in undergraduate education modules/mentoring students for teaching practice experience (WIL) |
| University of Stellenbosch (US) | Western Cape | Education | Teaching in undergraduate education |
| University of the Witwatersrand (Wits) | Gauteng | Humanities | No teaching in undergraduate education modules indicated |
| University of Zululand | KwaZulu-Natal | Education | Teaching in undergraduate modules |

Table 1 presents South African public universities with HPCSA-accredited master's programmes in educational psychology. These universities train educational psychologists, and the table shows their involvement in teacher education. Further investigation into the level of inclusion of educational psychology in teacher training programmes reveals that 29 higher education institutions in South Africa offer teacher education qualifications (see Table 2). Table 2 lists the universities in South Africa that offer undergraduate teacher education qualifications (public and private institutions).

Table 1 shows that there is limited access to master's programmes in educational psychology in South Africa. The information in Table 1 shows that there are not enough higher education institutions that offer educational psychology programmes to meet the demands of teacher training institutions shown in Table 2. Based on Tables 1 and 2, it can be argued that there is insufficient educational psychology contribution to the training of preservice teachers, and ultimately, there will not be enough educational psychologists to support the needs of learners in schools.

**Table 2.** List of universities that offer teacher education programmes in South Africa.

| University Offering Teacher Education Programmes | Province | Public | Private |
|---|---|---|---|
| University of South Africa (UNISA) | Gauteng | X | |
| University of the Western Cape | Western Cape | X | |
| University of the Witwatersrand | Gauteng | X | |
| University of Johannesburg | Gauteng | X | |
| Tshwane University of Technology | Gauteng | X | |
| Cape Peninsula University of Technology | Western Cape | X | |
| University of Cape Town | Western Cape | X | |
| Stellenbosch University | Western Cape | X | |
| Rhodes University | Eastern Cape | X | |
| Central University of Technology | Free State | X | |
| Vaal University of Technology | Gauteng | X | |
| University of Venda | Limpopo | X | |
| University of Zululand | KwaZulu-Natal | X | |
| University of KwaZulu-Natal | KwaZulu-Natal | X | |
| Durban University of Technology | KwaZulu-Natal | X | |
| North-West University (Mahikeng and Potchefstroom Campus) | North-West | X | |
| Nelson Mandela University | Eastern Cape | X | |
| University of Fort Hare | Eastern Cape | X | |
| Walter Sisulu University | Eastern Cape | X | |
| University of Pretoria | Gauteng | X | |
| University of Limpopo | Limpopo | X | |
| University of Mpumalanga | Mpumalanga | X | |
| Sol Plaatje University | Northern Cape | X | |
| University of the Free State | Free State | X | |
| Varsity College | Gauteng | | X |
| Mancosa | Gauteng | | X |
| Rosebank College | Gauteng | | X |
| Aros | Gauteng | | X |
| Akademia | Gauteng | | X |

It is difficult, if not impossible, to dissociate educational psychology from teacher training. The author of [2] (p. 6) declared that if the purpose of education is to promote change in the intellect, character and behaviour of people, psychology is the science that provides "thinkers and workers in the field of education (e.g., teachers) with knowledge of the material with which they work". Teacher education programmes are offered at most higher education institutions in South Africa. Table 2 lists 29 private and public universities. Based on the data from Tables 1 and 2, all 6 universities that offer HPCSA-accredited master's degree in educational psychology train teachers. However, the other 23 universities that train teachers do not have educational psychology programmes. Given the significant contribution of educational psychology to teacher training programmes, it is essential to motivate the curricula of preservice teachers trained at these 23 universities to focus on the three knowledge areas suggested by the framework for teacher education, namely knowledge of child development and learning assessment, inclusive education,

and support for learning problems. In this respect, [9] argues that teacher education programmes should include psychological topics like learning, development and assessment. Assessment for learning provides information on how to improve teaching and learning, while assessment of learning provides information on what and how much a learner has learned [31]. The author of [22] suggested that teacher education curricula should cover topics on how people learn, how to teach effectively, pedagogical content and language knowledge, and culture and community contexts.

Furthermore, it is concerning that only six universities offer educational psychology programmes that culminate into a master's degree in educational psychology accredited by the HPCSA. The spread of these universities across the provinces creates a lack of educational psychology programmes in other provinces of South Africa. Of these, three universities are in Gauteng, two are in KwaZulu-Natal and one is in the Western Cape (see Table 1). The study conducted by [32] reports how the training of educational psychologists prepares them to work in school contexts with limited resources. Existing statistics show that South Africa has 22,589 public schools, 12,684,886 learners in public schools, and 889 school circuits (Mweli); it can be assumed that each circuit has a minimum of 25 schools. Furthermore, the record shows that in public schools, psychologists are employed at the district level, and one psychologist can be responsible for a minimum of 30 to 70 schools, depending on how big the district is and how many circuits are in that district. Each circuit has a minimum of 8 schools and a maximum of 25. Again, South Africa has approximately 1672 registered educational psychologists [24]. Based on the above statistics, learners at the school level cannot access the support of an educational psychologist at every school; there are not enough educational psychologists to accommodate all learners in public schools. Ultimately, teacher education programmes must include in their curriculum courses and modules that will empower teachers to understand child development, learning and barriers to learning, as well as how to screen, assess and implement intervention programmes to support learners with learning barriers so that they can function independently in these areas at the school level due to the limited access to the support of an educational psychologist. The shortage of educational psychologists is dire, and the limited higher education training programmes (see Table 1) will not address the problem. Educational psychologists are sometimes referred to as school psychologists in the USA, which also reports a national shortage of school psychologists. This problem presents a global concern. In [33], it is reported that the shortage spans from practitioners and graduate programmes. The National Association of School Psychologist (NAPS) confirms a ratio of one educational psychologist to 1211/1500 learners (1:1211/1500), depending on the state. This situation is catastrophic.

The Department of Basic Education [13,21], through its White Paper on inclusive education, established district-based support teams to support all public schools in South Arica, with educational psychologists as part of this team. The primary function of district-based support teams is to "evaluate programmes, diagnose their effectiveness and suggest modifications, support teaching, learning and management, build the capacity of schools, recognise and address severe learning difficulties and accommodate a range of learning needs" [21]. This centralised process of support by multidisciplinary teams of professionals offers much-needed specialist services to public schools. The education specialists (four) who responded to the email reported that each school circuit, with approximately 25 schools, is allocated an educational psychologist; this could amount to thousands of learners for an allocated school. Thus, accessing and offering support to all learners is impossible. Ultimately, since all qualified educational psychologists are not employed in schools, they can support learners through work in private practice. This is confirmed by the authors of [24,34], who argue that most educational psychologists work in private practices focusing on psycho-educational assessments, parental guidance, and child psychotherapy. Considering the context of South Africa with a scarcity of educational psychologists to support learners in schools, there is a need to engage with higher education and basic education departments to assess how best the shortage of educational psychol-

ogy programmes in universities can be addressed and how the training of educational psychologist can be promoted for services in public schools. Finally, future research should focus on how educational psychologists are distributed and allocated to schools within the district-based support teams. There is a dearth of literature in these specialist areas to augment the current statistics on learners, schools and training of both preservice teachers and educational psychologists.

### 5. Conclusions and Recommendations

There is a shortage of educational psychology programmes that train preservice teachers in South African high education institutions. Likewise, few educational psychologists are trained in educational psychology programmes. To ensure that teachers are adequately trained in educational psychology, teacher education curricula should include theories of child development, learning, assessment and learner support so that teachers can be equipped to work independently in schools with limited or no support from educational psychologists (except for cases that require referral).

This study recommends that the HPCSA and the South African Department of Basic Education create opportunities to discuss areas of concern regarding the role of educational psychologists in schools. One possibility is to create opportunities for increasing the number of schools that can serve as internship sites where educational psychology interns may work for a year to alleviate the workload of educational psychologists in the Department of Basic Education. In this process, it will offer the needed learning support to learners in schools. Higher education institutions should create possibilities for educational psychology programmes in all teacher education institutions. If this is not possible, the curriculum for teacher education should equip teachers with educational psychology.

Given the scarcity of training programmes and related departments for educational psychology at higher education institutions, it is unrealistic to expect all teacher education institutions to have qualified educational psychologists involved in teacher training. However, the curriculum of preservice teachers can be strengthened to include educational psychology.

Finally, the future of educational psychology programmes and departments in South African higher education institutions requires urgent attention. Table 1 shows that higher education institutions in six South African provinces, including Northwest, the Eastern Cape, the Northern Cape, the Free State, Limpopo and Mpumalanga, do not train educational psychologists. However, a growing number of learners in schools require the support of educational psychologists. Furthermore, not all educational psychologists in South Africa are employed within the basic education school system.

**Funding:** This research received no external funding.

**Institutional Review Board Statement:** Not applicable.

**Informed Consent Statement:** Not applicable.

**Data Availability Statement:** A narrative literature review was conducted for this research. The email correspondence with the educational psychologists to clarify the second research question due to the dearth of literature will be made available should that be necessary.

**Conflicts of Interest:** The author declares no conflict of interest.

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
