# Peer review of "The Contribution of Educational Psychology to South African Preservice Teacher Training and Learner Support"

_education, doi:10.3390/educsci13101047_

Round 1

Reviewer 1 Report

The article requires major restructuring and improvement. Paper sections are not placed in the correct order (e.g., theoretical framework is positioned after Reserach Methods section).

Literature review should not include statistical data on schools and universities. Maybe the title literature review is not correct. Also, the authors are making conclusions about appropriateness of some solutions, problems etc., that type of statements do not belong to the literature review.  

The research method section is unclear. The authors should clearly state what type of narrative review are using in this study and not teach readers about different types of narrative reviews.

In the research method section, the authors are stating research questions. For clarity purposes the research questions should be in a separate section. After stating the questions, they explain the data collection procedure for the second research question, and even start with a partial answer to the second question. What about the first research question data collection procedure?

It is not necessary to state numerical data in the abstract.

Author Response

COVER LETTER (Summary of the changes made)

I thank the reviewer for the comments and suggestions to enhance the manuscript’s quality.

Manuscript title: The contribution of educational psychology in South African preservice teacher training and learner support

Kindly note that this Table of Corrections must be read with the corrected manuscript.

Comments: Reviewer 1

Corrections by author

Page No.

1.

Is the content succinctly described and

contextualized with respect to previous and

present theoretical background and empirical

research (if applicable) on the topic?-

Must be improved.

 The introduction, theoretical framework and literature review sections were rearranged, and the literature from the discussion sections was also moved to the literature review sections to strengthen the content and respond to this comment. 

 2-4 (1.2);

2.

Are all the cited references relevant to the

research?

Can be improve.

 I/we have added references to expand the discussion; these are indicated in purple in the manuscript's content. [Authors of 3; 19; 20; 28]

3.

Are the research design, questions, hypotheses

and methods clearly stated?

Must be improve.

Section 3 discusses research methods and questions for the study.

 page 5-6

  4.

Are the arguments and discussion of findings

coherent, balanced and compelling?

Must be improved

 I/we have separated the discussions and findings from the literature review to improve the argument based on the literature reviewed.

 Pages 7-9

  5.

For empirical research, are the results clearly

presented?

Must be improve.

The results presented in section 4 are to expand on the patterns identified from the literature reviewed -

 Pages 6-9

   6.

Is the article adequately referenced?

Can be improved.

 A total of 34 references were used in this manuscript which I think is to address the adequate referencing of relevant literature

See reference section

  7 

Are the conclusions thoroughly supported by the

results presented in the article or referenced in

secondary literature?

Must be improved.

 I/we have separated the findings and discussions where I have used literature to support the findings. The conclusion and recommendations are based on findings and we have not used any references in this section to ensure avoid repetition but to also clearly state what the recommendations are.

8

The article requires major restructuring and improvement. Paper sections are not placed in the correct order (e.g., theoretical framework is positioned after Research Methods section).

The manuscript was restructured. The literature review was moved to immediately after introduction to ensure continuity (see section 1.2).

The theoretical framework was move to after literature review (a few papers published in the Education Journal were studies to follow the appropriate structure followed).

2-4

9

Literature review should not include statistical data on schools and universities. Maybe the title literature review is not correct. Also, the authors are making conclusions about appropriateness of some solutions, problems etc., that type of statements do not belong to the literature review.

The restructuring let to the tables moved to the finding sections of the manuscript. Literature on findings was separated from the narrative literature review which was moved to literature review section. See sections 1.2 for literature review and 4 for findings and discussions.

p. 2-4;

p. 6-9

10

The research method section is unclear. The authors should clearly state what type of narrative review are using in this study and not teach readers about different types of narrative reviews.

Under research method, it is indicated that the general literature review is followed. The description of types of available narrative literature reviews were not removed, but an addition on what type this study is following was added.

p.5

11

In the research method section, the authors are stating research questions. For clarity purposes the research questions should be in a separate section. After stating the questions, they explain the data collection procedure for the second research question, and even start with a partial answer to the second question. What about the first research question data collection procedure?

Section 3.1 provide research questions and objectives with a separate heading. The findings are only discussed in the section 4 which focuses only on findings and discussions.

p.6

12

It is not necessary to state numerical data in the abstract.

The numerical data was removed from the abstract (it is only in the findings sections).

p.8

Reviewer 2 Report

The theme of the article is very important and relevant. First of all, I congratulate the authors for a well written work. In general, the article is written with clear and well-defined sections. It presents updated and appropriate references for the type of investigation. The title is descriptive and consistent with the abstract and the article. However, the abstract does not provide an overview of the work. Aspects regarding the methodology are missing, such as a way of analyzing data and presenting the main results. The keywords are consistent with the work. (Please Remove the numbers from the words). The introduction includes important and relevant studies that can support the research, but these are few, there is space, and this discussion can be expanded. Some parts of the text could be better discussed.

The objectives are not presented clearly, the research question is only presented in the methodology. I suggest that, when reviewing the literature, show the importance of the work and present the objectives of the research. The methodology is not detailed. How was the data analysis carried out? Please, provide more details on how the General literature review was carried out.

I would like to see more links between related literature and your results in the discussion section and explanations of the possible reasons for the results that were found.

I believe these aspects can be improved and the article resubmitted for consideration.

Author Response

COVER LETTER (Summary of the changes made)

I / we thank the reviewer for the comments and suggestions to enhance the manuscript’s quality.

Manuscript title: The contribution of educational psychology in South African preservice teacher training and learner support

Kindly note that this Table of Corrections must be read with the corrected manuscript.

Comments: Reviewer 2

Corrections by author

Page No.

1.

Is the content succinctly described and

contextualized with respect to previous and

present theoretical background and empirical

research (if applicable) on the topic?-

Can be improved.

 The introduction, theoretical framework and literature review sections were rearranged, and the literature from the discussion sections was also moved to the literature review sections to strengthen the content and respond to this comment. 

 2-4 (1.2);

2.

Are all the cited references relevant to the

research?

Can be improved.

 I / we have added references to expand the discussion; these are indicated in purple in the manuscript's content. [Authors of 3; 19; 20; 28]

3.

Are the research design, questions, hypotheses

and methods clearly stated?

Must be improved.

Section 3 discusses research methods and questions for the study.

 page 5-6

  4.

Are the arguments and discussion of findings

coherent, balanced and compelling?

Can be improved.

 I/we have separated the discussions and findings from the literature review to improve the argument based on the literature reviewed.

 Pages 7-9

  5.

For empirical research, are the results clearly

presented?

Must be improved.

The results presented in section 4 are to expand on the patterns identified from the literature reviewed -

 Pages 6-9

   6.

Is the article adequately referenced?

Can be improved.

 A total of 34 references were used in this manuscript which I think is to address the adequate referencing of relevant literature

See reference section

  7 

Are the conclusions thoroughly supported by the

results presented in the article or referenced in

secondary literature?

Must be improved.

 I/we have separated the findings and discussions where I have used literature to support the findings. The conclusion and recommendations are based on findings and we have not used any references in this section to ensure avoid repetition but to also clearly state what the recommendations are.

8.

The theme of the article is very important and relevant. First of all, I congratulate the authors for a well written work. In general, the article is written with clear and well-defined sections. It presents updated and appropriate references for the type of investigation. The title is descriptive and consistent with the abstract and the article.

Thank you very much

The abstract does not provide an overview

of the work. Aspects regarding the methodology are missing, such as a way of analyzing data and presenting the main results.

I /we have included methodology the following sentence in the abstract:

A narrative literature review and email request for unpublished documents from 4 educational psychologist were used as a method to collect literature to answer the research questions:

Abstract

The introduction includes important and relevant studies that can support the research, but these are few, there is space, and this discussion can be expanded. Some parts of the text could be better discussed.

Additional 3 pages were added to support introduction under literature review. The literature review was separated from the introduction to avoid a long introduction, see section 1.2

2-4

The objectives are not presented clearly, the research question is only presented in the

methodology. I suggest that, when reviewing the literature, show the importance of the work and present the objectives of the research. The methodology is not detailed. How was the data analysis carried out? Please, provide more details on how the General literature review was carried out.

The objectives are included in section 3.1 under the research methods.

Section 3 – under research methods, general literature review strategy used is discussed (additional reference [28] was included to support the initial one.

Literature analysis was indicated under section 3.

p.6

I would like to see more links between related literature and your results in the discussion section and explanations of the possible reasons for the results that were found.

I believe these aspects can be improved and the article resubmitted for consideration.

The literature was re-arranged to ensure that the findings and results section is based on the sections of literature reviewed in pages 1-4. The findings are discussed from literature reviewed and the inclusion of theoretical framework discussed in section 2.

p. 6-9

Round 2

Reviewer 1 Report

Thank you for the revisions.

Reviewer 2 Report

I congratulate the authors for the effort and the changes they made to the article. It is evident that the authors incorporated the suggested changes, leading to an enhancement in the article's quality. The article is now ready for publication in this new version.